# Impact of Ni Content on the Electrochemical Performance of the Co-Free, Li and Mn-Rich Layered Cathode Materials

**Gongshin Qi \*, Jiazhi Hu, Michael Balogh, Lei Wang, Devendrasinh Darbar \*** and **Wei Li**

General Motors Global Research and Development Center, Warren, MI 48092, USA
\* Correspondence: gongshin.qi@gm.com (G.Q.); devendrasinh.darbar@gm.com (D.D.)

**Abstract:** Li and Mn-rich layered cathode (LLC) materials show great potential as the next generation cathode materials because of their high, practical and achievable specific capacity of ~250 mAh/g, thermal stability and lower raw material cost. However, LLC materials suffer from degradation of specific capacity, voltage fading due to phase transformation upon cycling and transition-metal dissolution, which presents a significant barrier for commercialization. Here, we report the effects of Ni content on the electrochemical performance, structural and thermal stability of a series of Co-free, LLC materials ($Li_{1.2}Ni_xMn_{0.8-x}O_2$, x = 0.12, 0.18, 0.24, 0.30 and 0.36) synthesized via a sol-gel method. Our study shows that the structure of the material as well as the electrochemical and thermal stability properties of the LLC materials are strongly dependent on the Ni or Mn content. An increase in the Ni to Mn ratio results in an increase in the average discharge voltage and capacity, as well as improved structural stability but decreased thermal stability.

**Keywords:** Li and Mn-rich layered cathode (LLC); sol-gel method; differential scanning calorimetry (DSC); cyclic voltammetry (CV)

## 1. Introduction

Lithium-ion batteries (LIBs) have become the most important secondary/rechargeable battery technology for portable electronic devices; and more recently for electric vehicles (EVs) [1–3]. Extensive research has been conducted on high-capacity electrode materials for LIBs due to the emerging demands for high energy density batteries in EVs [3–5]. Conventional cationic redox based cathode materials are $LiNi_{0.80}Mn_{0.10}Co_{0.10}O_2$ (NMC811), $LiNi_{0.5}Mn_{1.5}O_4$ and $LiFePO_4$, whereas Li and Mn-rich layered cathode (LLC) materials, containing both cationic and anionic redox processes, show great potential as the next generation cathode materials due to their higher theoretical specific capacities (up to 350 mAh/g) and lower raw material costs [6–10].

LLC materials have the same layered structure as conventional NMC811 except that the diffracted peaks in the XRD pattern between 20 and 30° uniquely belong to the monoclinic $Li_2MnO_3$ phase [9,11]. The integrated intensity ratio of the $I_{(003)}/I_{(104)}$ peak has been used as a key factor to reflect the extent of cation disorder [12] and, thus, predict the electrochemical properties of LLC materials [11–13].

The LLC materials suffer from voltage and capacity fading, transition-metal dissolution and rate performance limitations [6–10,14–16]. Many of these limitations originate from material structural changes with cycling. Therefore, stabilizing the LLC structure is critical to increasing the energy density retention rate, a key metric for the commercialization of this material. Kim et al. [17], Croy et al. [18] and Hy et al. [19] suggested that the nickel-manganese interaction within these materials plays a strong role in the stabilization of LLC materials. Compared to the other transition metals, such as $Fe^{2+}$ and $Cu^{2+}$. $Ni^{2+}$ provides better electrochemical/structural stability and phase purity for oxide-based cathodes [20]. It is well known that Ni provides a high capacity but poor thermal stability,

while Mn maintains an outstanding cycle life and enhances safety. Hence, the intrinsic properties of LLC materials could strongly depend on the relative amounts of Ni and Mn.

Here, we report a series of cobalt-free, Li and Mn enriched materials with a chemical formulation of $Li_{1.2}Ni_xMn_{0.8-x}O_2$ (x = 0.12, 0.18, 0.24, 0.30 and 0.36) prepared by a sol-gel method, and a detailed study on the effects of Ni content on their first cycle efficiency, charge and discharge capacities and thermal stability. Our findings could provide deep insights to guide the material design and commercial application of LLC materials in the future.

## 2. Experimental

### 2.1. Synthesis of LLC Materials

Lithium-rich layered cathode (LLC) materials were synthesized using the precursors of $Mn(NO_3)_2$ (98% purity, Alfa Aesar, Haverhill, MA, USA), $Ni(NO_3)_2.6H_2O$ (98% purity, Alfa Aesar, Haverhill, MA, USA) and $LiNO_3$ (99% purity, Alfa Aesar, Haverhill, MA, USA) at the designated compositions. Composition of the materials synthesized are $Li_{1.2}Ni_{0.12}Mn_{0.68}O_2$ (LLC−12), $Li_{1.2}Ni_{0.18}Mn_{0.62}O_2$ (LLC−18), $Li_{1.2}Ni_{0.24}Mn_{0.56}O_2$ (LLC−24), $Li_{1.2}Ni_{0.30}Mn_{0.50}O_2$ (LLC−30) and $Li_{1.2}Ni_{0.36}Mn_{0.44}O_2$ (LLC−36). The metal salts along with citric acid (from Sigma-Aldrich, 99.5%, molar ratio of citric acid/total metals = 1.1) were dissolved in water with continuous stirring to form a uniform precursor solution. The solution was evaporated slowly by heating to approximately 90 °C to produce a dry powder. The dry powder undergoes a three step: heating process, at 300 °C for 5 h (2 °C/min) in air for the decomposition of nitrates, followed by 700 °C for 5 h (2 °C/min) and at 900 °C for 20 h (2 °C/min) in air and cooled to room temperature.

### 2.2. Slurry Mixing and Electrode Coating

The LLC cathode electrodes are composed of 80 wt.% active material, 10 wt.% carbon black and 10 wt.% binder. Super P was used as the conductive filler and polyvinylidene fluoride (PVDF) as the binder. The electrode slurries were coated on Al foils and dried under vacuum at 60 °C overnight.

### 2.3. Cell Fabrication and Testing

Coin-type half-cells (CR2032) were fabricated inside an Ar-filled glovebox (<1 ppm oxygen and <1 ppm moisture) using LLC as the working electrodes and lithium (Li) chips (from MTI) as the counter electrodes. The electrolyte was 1.2 M $LiPF_6$ dissolved in 20 wt.% fluoroethylene carbonate (FEC), 80 wt.% dimethyl carbonate (DMC) and the separator was Celgard 2325. The mass loading of the cathode active material was approximately 9 mg/cm$^2$.

The cells were tested by galvanostatic charge/discharge cycling on a battery testing system (LAND, Wuhan, China) at 25 °C. For the cyclic performance assessment, two C/20 charge and discharge formation cycles were first applied in constant current mode. After the two formation cycles, the currents for charging and discharging were both changed to C/5 for cycle life testing. The voltage range between the electrodes was 2.0–4.7 V for the formation cycles and 2.0–4.6 V for the cycle life test.

### 2.4. Characterization

Scanning electron microscopy (SEM) data were collected using a Hitachi S-4800 field emission SEM operating at 5 kV. Powder samples were prepared by sprinkling the sample lightly across double-sided conductive carbon tape adhered to a sample stub.

Powder X-ray diffraction (XRD) data were collected using a Bruker D8 Advanced X-ray Diffractometer using copper k-alpha radiation generated with a copper x-ray tube. Rietveld refinements were performed using Topas software from Bruker. All refinements optimized the $R\bar{3}m$ structure model with full occupancy of $O^{-2}$ on the 6c site and the cations on the 3a and 3b sites. The cation ratios were fixed to the theoretical values of each sample while maintaining charge balance. All refinements optimized cation mixing, the 6c

site position and lattice parameters. Cation mixing was defined as the percentage of the 3a site occupied by the combined Mn and Ni occupancies.

For analysis by differential scanning calorimetry (DSC), the 2032 coin-type half-cells were fully charged to 4.7 V (100% SOC) and opened in an Ar-filled glove box. The electrodes were rinsed in dimethyl carbonate (DMC). Once dried, the samples were weighed and then transported to the DSC equipment. The samples were briefly exposed to air during the short time of loading the sample in the instrument. The measurements were carried out under argon up to 500 °C at a ramping rate of 5 °C/min in a DSC 200 Differential Scanning Calorimeter (NETZSH, Selb, Germany).

## 3. Results and Discussion

### 3.1. Cathode Active Material Structure and Morphology

Crystallographic evaluations were performed on the sol-gel synthesized LLC−12, LLC−18, LLC−24, LLC−30 and LLC−36 using XRD pattern shown in Figure 1a,b. All the LLCs cathode materials show that the diffraction peaks belong to the $R\bar{3}m$ space group of the rhombohedral $\alpha$-NaFeO$_2$ structure and the weak superlattice diffraction reflections (20–22.5°) ascribed to the monoclinic Li$_2$MnO$_3$ phase (C2/m space group) [6,21–23]. LLC−12 and LLC−18 show that an additional peak at ~36° corresponds to the spinel phase (*Fd3m*). Rietveld refinements were performed on the XRD pattern on all LLC materials, and Table 1 shows the unit cell lattice parameters and cation mixing. Unit cell volume and cation mixing increase with the increase in the Ni content, except for the LLC−36 cathode. This is because of larger ionic radius of Ni$^{2+}$ (0.69 Å) compared to Mn$^{4+}$ (0.53 Å). For LLC−36, the cation mixing is reduced to 3.2%, assuming there is a partial Ni$^{2+}$ to Ni$^{3+}$ conversion. The actual compositions of Li, Mn and Ni in the synthesized materials are shown in Table 2, analyzed by ICP analysis. Scanning electron microscopy (SEM) images of the LLCs with increasing Ni content are displayed in Figure 2. The particle size increasing with Ni content has a particle size distribution in the range of 200–600 nm. These results show that the Ni and Mn contents influence the microstructures of Li-rich layered cathode materials.

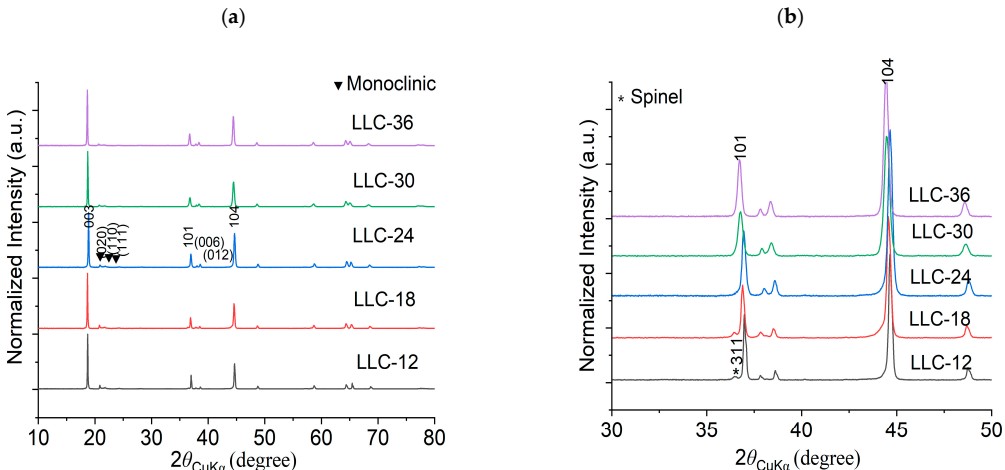

**Figure 1.** X-ray Diffraction patterns for all the LLCs powder samples (**a**) 2θ: 10–80° and (**b**) 2θ: 30–50°.

**Table 1.** Unit cell parameters of as-prepared LMR samples.

| Samples | R-3m Phase | | | | Fd3m Spinel Phase | |
| --- | --- | --- | --- | --- | --- | --- |
| | a (Å) | c (Å) | Vol (Å^3) | TM Mixing | a (Å) | Vol (Å^3) |
| LLC−12 | 2.8537 | 14.293 | 100.80 | <0.1% | 8.1906 | 549.47 |
| LLC−18 | 2.8584 | 14.280 | 101.04 | 1.5% | 8.1948 | 550.31 |
| LLC−24 | 2.8672 | 14.273 | 101.62 | 3.2% | Not observed | |
| LLC−30 | 2.8721 | 14.290 | 102.09 | 4.3% | Not observed | |
| LLC−36 | 2.8682 | 14.268 | 101.65 | 3.2% | Not observed | |

**Table 2.** Composition of Li, Ni and Mn for the as-prepared samples.

| Samples | Composition Measured (%) | | | Theoretic Composition (%) | | |
|---|---|---|---|---|---|---|
| | Li | Ni | Mn | Li | Ni | Mn |
| LLC−12 | 9.78 | 7.88 | 44.2 | 9.83 | 8.3 | 44.1 |
| LLC−18 | 9.84 | 12.3 | 41.6 | 9.80 | 12.4 | 40.1 |
| LLC−24 | 9.47 | 15.9 | 40.1 | 9.78 | 16.5 | 36.1 |
| LLC−30 | 9.35 | 21.6 | 33.1 | 9.75 | 20.6 | 32.2 |
| LLC−36 | 9.62 | 25.7 | 29.6 | 9.73 | 24.7 | 28.2 |

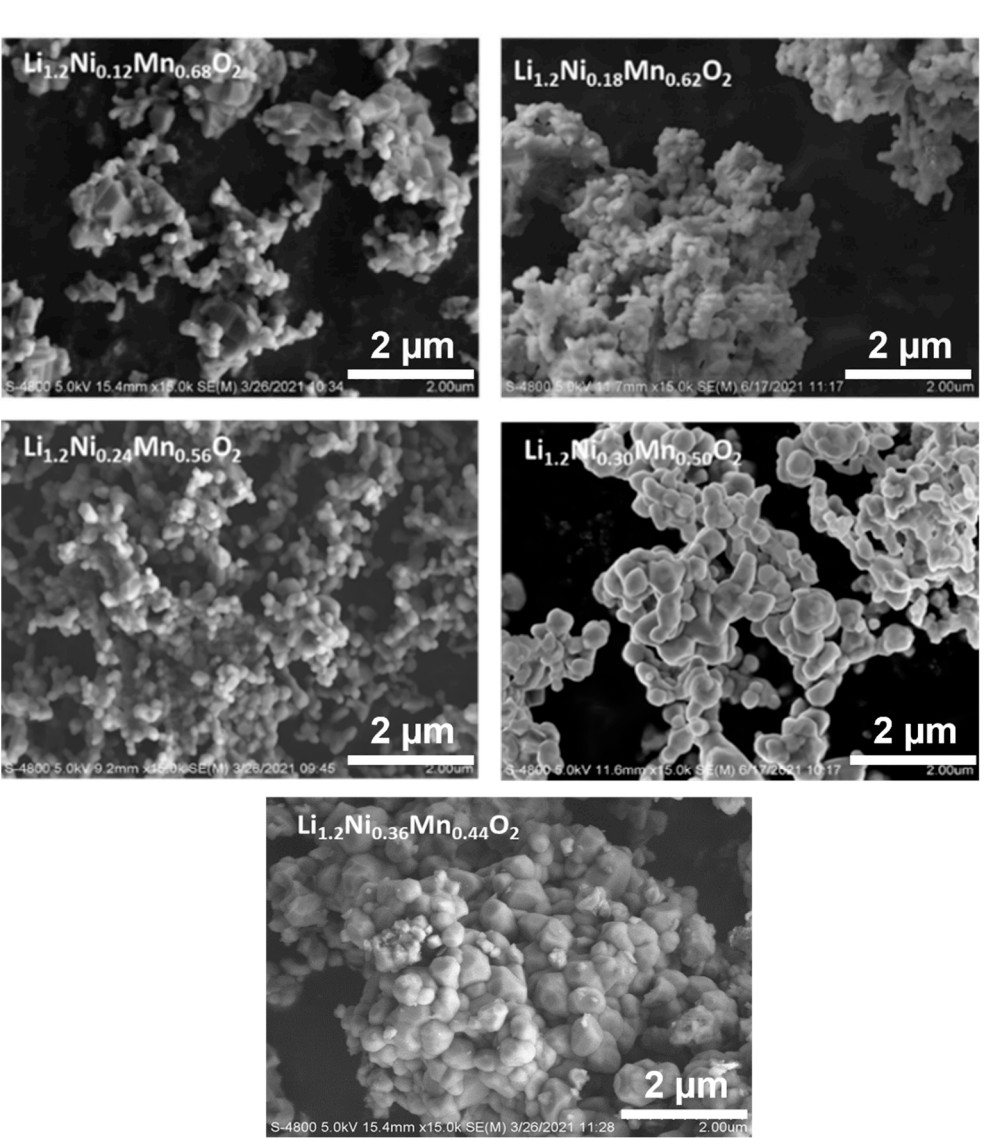

**Figure 2.** SEM image of all as-prepared LLCs powder samples.

### 3.2. Electrochemical Evaluation for LLCs

Figure 3 shows 1st-cycle voltage profiles and dQ/dV plots for all LLCs, which cycle between 2.0–4.7 V vs. Li/Li$^+$ at a rate of C/20 (1 C = 210 mAh/g). In Figure 3a, three regions are shown during the charging process: the slope region with voltage below 4.5 V corresponding to the Ni$^{2+}$ oxidation to Ni$^{4+}$, the plateau region with voltage at approximately 4.5 V attributed to the activation of the Li$_2$MnO$_3$ phase and the oxidation of O$^{2-}$ to peroxo (O$_2$)$^{n-}$ dimer species with voltage above 4.5 V [24–29]. The nature of the O$^{2-}$ oxidation/reduction mechanism remains controversial, whereas consensus has been reached that the redox of O$^{2-}$ contributed to the enhanced capacity for LLC

materials, in addition to the conventional cationic redox. The charge profiles (Figure 3b) are characterized by a strong anodic peak at approximately 4.53 V, except for LLC−12 and LLC−18, which have anodic peaks at 4.64 V and 4.57 V, respectively. A weak and broad anodic peak at approximately 3.78 V, attributed to the $Ni^{2+}$ oxidation to $Ni^{3+}$ and $Ni^{4+}$ [25–29] is observed on all the samples. The LLC−12 and LLC−18 samples not only showed lower discharge capacities, but also lower discharge voltages (Figure 3c) compared to the other three samples. Three main reduction peaks/regions are observed (see Figure 3d). The first one located at 4.33 V corresponds to $Ni^{4+}$ reduction to $Ni^{3+}$, the second one located at approximately 3.76 V is assigned to $Ni^{3+}$ reduction to $Ni^{2+}$, and the third one located at 3.35 V corresponds to $Mn^{4+}$ reduction to $Mn^{3+}$ of the activated $MnO_2$ phase [29]. In addition to the three main peaks mentioned above, another weak peak at 2.77 V observed for the samples LLC−12, LLC−18 and LLC−24 can be assigned to the $Mn^{4+}$ reduction to $Mn^{3+}$ of the spinel phase.

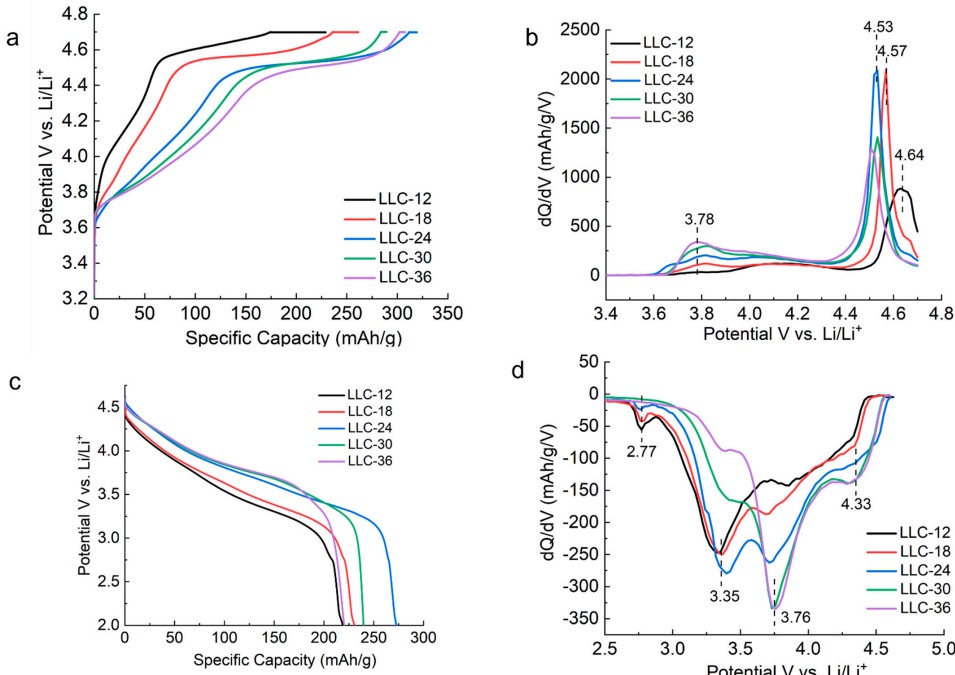

**Figure 3.** (**a**) First−cycle specific charge capacity vs. voltage curve. (**b**) Differential capacity (dQ/dV) plots for 1st cycle charging. (**c**) First–cycle specific discharge capacity vs. voltage curve and (**d**) differential capacity (dQ/dV) plots for 1st cycle discharge for all the LLC electrodes. Cells were cycled between 2.0 and 4.7 V at a C/20 rate.

Table 3 summarizes the first-cycle specific capacity and Coulombic efficiency for all the samples. The contributions of $Ni^{2+}$ oxidation and $Li_2MnO_3$ activation to the capacity are calculated based on the voltage profile. The capacity from $Ni^{2+}$ oxidation increases while that from $Li_2MnO_3$ activation decreases with Ni content. By combining these two factors, the LLC−24 sample shows the highest capacity. However, the first cycle Coulombic efficiency decreases as the Ni content increases, which attribute to the increase in cation mixing. The Coulombic efficiencies of LLC−12 and LLC−18 samples are higher than LLC−24, LLC−30 and LLC−36 because better Li-ion diffusion pathway formed from spinel structure (See Figure 4).

**Table 3.** Specific charge and discharge capacities and Coulombic efficiencies of LLCs.

| Samples | 1st Cycle Specific Charging Capacities from $Ni^{2+}$ Oxidation (mAh/g) | 1st Cycle Specific Charging Capacities from $Li_2MnO_3$ (mAh/g) | 1st Cycle Total Specific Charging Capacities (mAh/g) | 1st Cycle Specific Discharging Capacities (mAh/g) | 1st Cycle Coulombic Efficiencies (%) | Specific Discharging Capacities (mAh/g) after 20th Cycle |
|---------|------|------|------|------|------|------|
| LLC−12 | 61 | 174 | 235 | 224 | 95.3 | 191 |
| LLC−18 | 87 | 184 | 271 | 239 | 88.2 | 181 |
| LLC−24 | 158 | 161 | 319 | 272 | 85.2 | 210 |
| LLC−30 | 172 | 118 | 290 | 240 | 82.8 | 208 |
| LLC−36 | 210 | 84 | 294 | 217 | 73.8 | 187 |

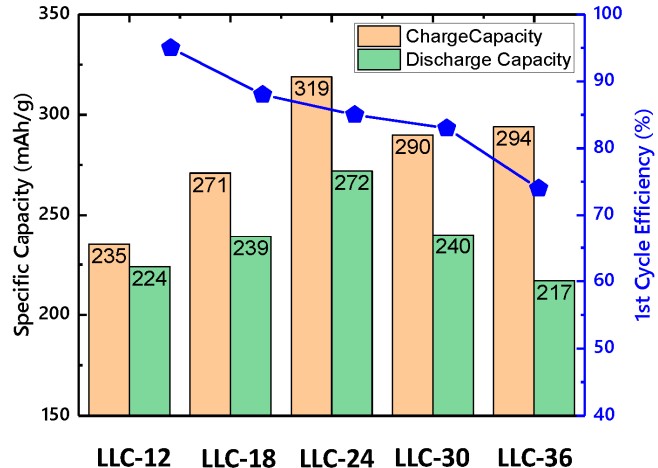

**Figure 4.** Specific capacity and 1st-cycle coulombic efficiency for LLCs electrode.

As shown in Figure 5a, specific capacity vs. cycle number after 20 cycles, the discharge capacities for LLC−12, LLC−18, LLC−24, LLC−30 and LLC−36 are 199, 181, 210, 208 and 187 mAh/g, respectively. Figure 5b shows the average discharge voltage profiles as a function of the cycle numbers. Clearly, the average discharge voltage increases with the increase in Ni content, and the LLC−36 sample reaches the lowest voltage decay rate and the highest average discharge voltage. Our results are consistent with the report from Shi et al. [30,31] where they demonstrated that Ni acts as stabilizing ions to inhibit the Jahn-Teller effect of $Mn^{3+}$ ions, supporting the layered structure as a pillar. Ni ions can migrate between the transition metal layer and the Li ion layer, which avoids the formation of the spinel-like structure and mitigates the voltage decay within certain Mn/Ni ratios.

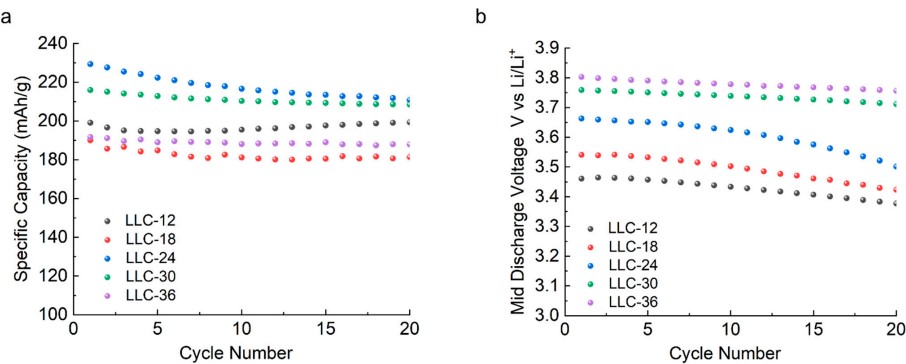

**Figure 5.** (**a**) Specific discharge capacity and (**b**) average discharge voltage profiles of the investigated samples as a function of cycle numbers in a voltage range of 2.0–4.6 V under a C/5 rate.

Cyclic voltammetry (CV) was performed on all LLCs to study the oxidation and reduction in the cathode material during cycling. Figure 6 plots the specific current as a function of voltage for the first six cycles of each sample. The reaction peaks at approximately 3.25 V are related to the oxidation of $Mn^{3+}$ to $Mn^{4+}$ from the spinel phase in LLC−12, LLC−18 and LLC−24. During the first charging process, all the LLCs samples showed oxidation peak at approximately 4.70 V, which originates from the Li ion migration from the transition metal oxide layer and $Li_2MnO_3$ activation with the oxidation of $O^{2-}$ to $O^-$ or $O_2$ gas phase [32]. For the discharging process, the lower voltage peak located at approximately 2.70 V for LLC−12, LLC−18 and LLC−24 samples corresponds to the $Mn^{4+}$ reduction to $Mn^{3+}$ of the spinel structure, which is consistent with the results in Figure 6. The reduction peaks between 3.0 and 3.5 V result from the reduction of $Mn^{4+}$ to $Mn^{3+}$ of the spinel-like structure after $Li_2MnO_3$ activation [33], and the peak intensities decrease as the Ni content increases, implying that the reversible electrochemical activity of Mn ions decreases and eventually disappears when the Ni content reaches 0.36. The peaks located at approximately 4.2 V and 3.8 V were caused by the $Ni^{4+}$ reduction to $Ni^{3+}$ and $Ni^{3+}$ reduction to $Ni^{2+}$, respectively [34].

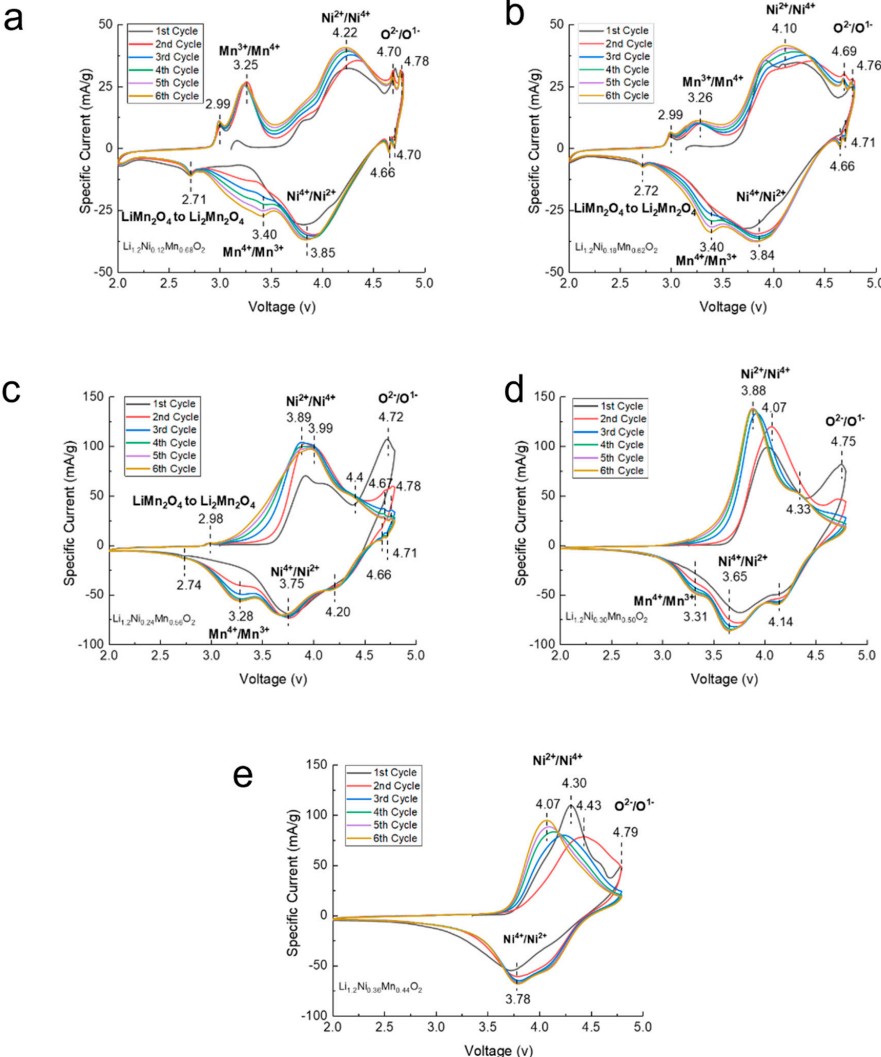

**Figure 6.** Cyclic voltammetry (CV) profiles of the first six cycles at a scanning rate of 0.1 mV/s for (**a**) LLC−12, (**b**) LLC−18, (**c**) LLC−24, (**d**) LLC−30 and (**e**) LLC−36.

After the first cycle, the most significant change in the CV profiles for the subsequent cycles is that the reduction peaks located at approximately 3.0–3.5 V gradually increase

for the LLC−12, LLC−18, LLC−24 and LLC−30 samples, indicating a phase transformation from the layered structure to the spinel-like structure. However, the intensity of the $Mn^{4+}/Mn^{3+}$ reduction peaks decreases as the Ni content increases. No $Mn^{4+}/Mn^{3+}$ reduction peak is observed on the LLC−36 sample, indicating that the spinel-like structure has not been formed.

*3.3. Thermal Stability of LMR Electrodes with Different Ni Contents*

The thermal stability of the delithiated LMR electrodes (without electrolyte) was evaluated by differential scanning calorimetry (DSC). Figure 7 shows the corresponding DSC testing results. Two exothermic peaks are observed on each fully delithiated electrode (100% SOC), and the exothermic reaction peak gradually shifts to lower temperatures as the Ni content increases, except for the LLC−30 electrode. It has been reported that the improved thermal stability of NMC-based cathodes with lower Ni content can be attributed to their more stable $Mn^{4+}$ content. The lower temperature peak is assigned to the decomposition of the cathode active materials, whereas the higher temperature peak is related to the decomposition of the binder and/or the reduction in the active material by the carbon black [35]. Table 4 lists the peak temperatures of the two exothermic reactions and the energy released per gram of the electrodes. The total heat generated increases with increasing Ni content, except for the LLC−24 sample, which generates the highest total heat among the five samples.

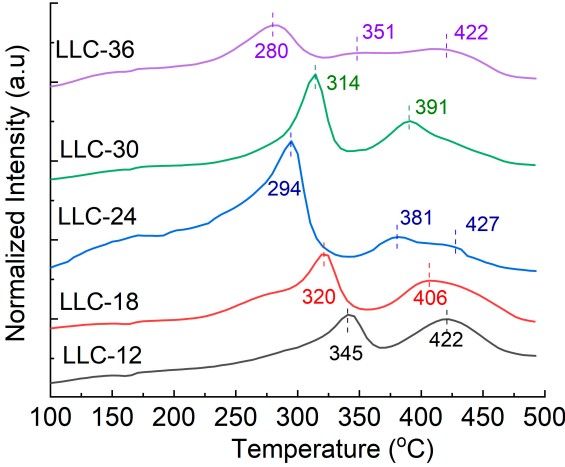

**Figure 7.** DSC results of all delithiated LLCs electrodes.

**Table 4.** Summary of the peak temperatures and heat generated.

| Samples | 1st Exothermic Peak Temperature/°C | 2nd Exothermic Peak Temperature/°C | Total Heat Released (J/mg) |
|---|---|---|---|
| LLC−12 | 345 | 422 | 0.89 |
| LLC−18 | 320 | 406 | 1.11 |
| LLC−24 | 294 | 381/427 | 1.77 |
| LLC−30 | 314 | 391 | 1.14 |
| LLC−36 | 280 | 351/422 | 1.23 |

## 4. Conclusions

This work has shown that the relative content of Ni and Mn in the Co-free LLC materials prepared by a sol-gel method, $Li_{1.2}Ni_xMn_{0.8-x}O_2$ (x = 0.12, 0.24, 0.30 and 0.36), has a significant impact on the material structure, electrochemical performance and thermal stability. Generally, as the Ni content increases, the primary particle size increases within a range of 200–600 nm. XRD results indicate that LLCs with higher Mn content (x = 0.12 and 0.18) show a layered (*R3m* and *C2/m*) and spinel (*Fd3m*) heterostructure, and the fraction

of the spinel phase decreases as the Ni content increases. Only one layered structure was observed on the samples with lower Mn contents (x = 0.24, 0.30 and 0.36). Mn or Ni content demonstrates a significant impact on the specific capacity, reaching the highest capacity when x = 0.24 or Mn/Ni mol ratio at 7:3. As the Ni content increases, the sample shows decreasing first-cycle Coulombic efficiency, but increasing average discharge voltage, higher electrochemical stability, and faster Li ion diffusivity. The LLC−12 sample exhibits the best thermal stability among the tested samples; however, its specific energy density is limited due to the lowest average discharge voltage. The LLC−36 sample, which contains the highest Ni content, exhibits the worst thermal stability but a relatively higher specific energy density due to the highest average discharge voltage. The highest specific energy density was achieved on the LLC−24 sample with decent thermal stability. Optimal Ni content is needed in LLC for optimal particle size and better phase homogeneity between spinel, monoclinic and layered, which attributes to better electrochemical performance as we observed for LLC−24 cathode material.

**Author Contributions:** G.Q.: Conceptualization, formal analysis and writing—original draft; J.H.: SEM, cycle data collection and plotting; M.B.: XRD and analysis; L.W.: DSC and CV test; D.D.: Formal analysis, writing—original draft; W.L.: Writing, review and editing. All authors have read and agreed to the published version of the manuscript.

**Funding:** This research received no external funding.

**Institutional Review Board Statement:** Not applicable.

**Informed Consent Statement:** Not applicable.

**Data Availability Statement:** Not applicable.

**Acknowledgments:** The authors would like to thank James Salvador for the DSC testing, and Shubha Nageswaran for CV test result discussion.

**Conflicts of Interest:** The authors declare no conflict of interest.

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
