# Peer review of "Impact of Ni Content on the Electrochemical Performance of the Co-Free, Li and Mn-Rich Layered Cathode Materials"

_2673-3293, doi:10.3390/electrochem4010002_

Round 1

Reviewer 1 Report

This work by Gongshin Qi Jiazhi Hu and coworkers addresses the electrochemical performance of layered cathode materials characterized by being Co-free as well as Li and Mn-rich, focusing on the Impact of the Ni content.

The topic is of extreme relevance cause it may impact the market of battery technology  with possible applications both in portable electronics and electric vehicles .

The authors investigate a series of cobalt-free, Li and Mn enriched samples synthesized  by a sol-gel method and carry on the study on the effects of Ni content on several figures of merit, namely the first cycle efficiency, the  charge/discharge capacities, the thermal stability.

Results are significant and well described and might drive future material design .

I think that this a is a solid work, suitable for the journal, and might be publishable in the present form.

Author Response

We thank the reviewers for their feedback and comments which strengthen the manuscript.  Here we provide our detailed responses to the reviewers’ comments and suggestions along with the revised manuscript.

List of changes made in the revised manuscript are below. All changes to the revised manuscript are indicated by red text:

  1. Address the reviewers’ comments throughout the main text.
  2. Revised/added discussion and polished grammar throughout the main text.

Reviewer 1

This work by Gongshin Qi Jiazhi Hu and coworkers addresses the electrochemical performance of layered cathode materials characterized by being Co-free as well as Li and Mn-rich, focusing on the Impact of the Ni content.

The topic is of extreme relevance cause it may impact the market of battery technology  with possible applications both in portable electronics and electric vehicles .

The authors investigate a series of cobalt-free, Li and Mn enriched samples synthesized  by a sol-gel method and carry on the study on the effects of Ni content on several figures of merit, namely the first cycle efficiency, the  charge/discharge capacities, the thermal stability.

Results are significant and well described and might drive future material design .

I think that this a is a solid work, suitable for the journal, and might be publishable in the present form.

Comments

We would like to thank the author for reviewing the manuscript and provide the valuable suggestions. 

Reviewer 2 Report

The manuscript by Hu et al entitled "Impact of Ni Content on the Electrochemical Performance of the Co-free, Li and Mn-rich Layered Cathode Materials"  is aimed at the analysis of the role of the Ni/Mn ratio to the electrochemical characteristics of Li-rich layered oxide cathode materials.

The topicality of the object of research is high due to the objective need to elucidate the role of the synthesis in obtaining the structurally stable cathode material befitting the current requirements to the cathode materials with high energy and power density, rate capability and increased safety. However, the content of the manuscript does not allow to use the results obtained in this study for the conclusions on the precise impact of Ni/Mn ratio on the key electrochemical characteristics of the synthesized compounds.

Unfortunately, the experiment has serious flaws impedient to this article be published in its current form.

The main  shortcomings are the follows:

1. The galvanostatic test have to be performed with the increased number of cycles: 20 cycles are not enough. 

2. The paragraph with the experimental details should be improved, some details were not given while others cause concerns.

3. Methods of characterization that are able to support the conclusions obtained in the experiment are needed (e.g., XPS, STEM). The presented results should be re-visited as there are some imperfections can be found: please, verify the Figures with XRD. The SEM images were obtained with the interval of four months, thus, it is unclear if some compounds were re-synthesized or the samples were stored. In the latter case,  it should be recommended to verify if the electrochemical characteristics of the samples are still the same.

3. Some results are controversial to the abstract.

4.   Some sentences should be explained:

"For LLC-36,the cation mixing is reduced to 3.2% assuming there is a partial Ni2+ to Ni3+"

"formation of amorphous compound" (please, see the text)

"syrupy mass" (I would recommend you to replace it with the conventional to sol-gel synthesis thermonology)

Author Response

Response to Reviewers (Electrochem-2112193)

We thank the reviewers for their feedback and comments which strengthen the manuscript.  Here we provide our detailed responses to the reviewers’ comments and suggestions along with the revised manuscript.

List of changes made in the revised manuscript are below. All changes to the revised manuscript are indicated by red text:

  1. Address the reviewers’ comments throughout the main text.
  2. Revised/added discussion and polished grammar throughout the main text.

Reviewer 3 Report

The paper by Qi et al. investigates the influence of Ni content on the electrochemical performance, structural and thermal stabilities of Li and Mn rich layered cathode (LCC). The prepared Li1.2NixMn0.8-xO2 has been characterized by the authors using various techniques, such as SEM, XRD and DSC. The authors found that layered structrures are only found at lower contents and that the specific capacity of the Li1.2NixMn0.8-xO2 depends on the Mn/Ni ratio. Overall, the paper provides useful insights into the effect of Ni on the structural and thermal stabilities and specific capacity of Li1.2NixMn0.8-xO2. This work is therefore suitable for publication in this journal after addressing the following comments:

1. In Figure 1a and b, the unit of 2 theta should be provided in the x-axis labels.

2. In Figure 1b, the label "b" seems to be cut off. Please fix this.

3. The heating rate(s) used for the synthesis of LLC materials should be specified in the Experimental section.

4. In Figure 3c, the bracket symbol for mAh/g is missing. Please keep consistency as Figure 3a.

5. A Table comparing the specific capacity of the optimum LLC (LLC-24) against other reported cathode materials need to be provided in this paper.

6. In the Conclusion section, the main reason behind the high capacity of LLC-24 compared to other LLC cathode materials needs to be summarized.

7. In the Experimental section, all the chemicals used and their purity should be supplied to allow for reproduction by readers.

8. In Figure 2, the original scale bars of the SEM images can be removed and replaced with redrawn scale bars.

9. In the Introduction section, the authors can explain more about the motivation behind the selection of Ni and not other transition metals.

10. Some spelling errors are observed. Please double check the spelling during revision stage.

Author Response

(The authors gave the same response as above.)

Round 2

Reviewer 2 Report

The topicality of the object of research is high due to the objective need to elucidate the role of the synthesis in obtaining the structurally stable cathode material befitting the current requirements to the cathode materials with high energy and power density, rate capability and increased safety. However, the content of the manuscript does not allow to use the results obtained in this study for the conclusions on the precise impact of Ni/Mn ratio on the key electrochemical characteristics of the synthesized compounds. The experimental part should be extended, the additional experiments are needed as the current one have some flows.